# Increased sensitivity of malaria parasites to common antimalaria drugs after the introduction of artemether-lumefantrine: Implication of policy change and implementation of more effective drugs in fight against malaria

Winnie Okore[1,2], Collins Ouma[2], Raphael O. Okoth[1], Redemptah Yeda[1], Luicer O. Ingasia[3], Edwin W. Mwakio[1], Douglas O. Ochora[1,4], Duncan M. Wakoli[1,5], Joseph G. Amwoma[1,6], Gladys C. Chemwor[1], Jackline A. Juma[1], Charles O. Okudo[1], Agnes C. Cheruiyot[1], Benjamin H. Opot[1], Dennis Juma[1], Timothy E. Egbo[7], Ben Andagalu[1], Amanda Roth[8], Edwin Kamau[9], Hoseah M. Akala[1]*

1 Department of Emerging and Infectious Diseases (DEID), United States Army Medical Research Directorate-Africa (USAMRD-A), Kenya Medical Research Institute (KEMRI)/Walter Reed Project (WRP), Kisumu, Kenya, 2 Department of Biomedical Sciences and Technology, Maseno University, Kisumu, Kenya, 3 Sydney Brenner Institute of Molecular Biosciences, University of Witwatersrand, Johannesburg, South Africa, 4 Department of Biological Sciences, Kisii University, Kisii, Kenya, 5 Department of Biochemistry and Molecular Biology, Egerton University, Njoro, Kenya, 6 Department of Biological Sciences, University of Embu, Embu, Kenya, 7 United States Army Medical Research Directorate-Africa (USAMRD-A), Kisumu, Kenya, 8 Medical Communications for Combat Casualty Care, Fort Detrick, Maryland, United States of America, 9 Department of Pathology and Area Laboratory Services, Tripler Army Medical Center, Honolulu, Honolulu, United States of America

* hoseaakala@yahoo.com, Hosea.akala@usamru-k.org

## Abstract

Single nucleotide polymorphisms (SNPs) in the *Plasmodium falciparum multi-drug resistance protein 1* (*Pfmrp1*) gene have previously been reported to confer resistance to Artemisinin-based Combination Therapies (ACTs) in Southeast Asia. A total of 300 samples collected from six sites between 2008 and 2019 under an ongoing malaria drug sensitivity patterns in Kenya study were evaluated for the presence of SNPs at *Pfmrp1* gene codons: H191Y, S437A, I876V, and F1390I using the Agena MassARRAY® platform. Each isolate was further tested against artemisinin (ART), lumefantrine (LU), amodiaquine (AQ), mefloquine (MQ), quinine (QN), and chloroquine (CQ) using malaria the SYBR Green I-based method to determine their *in vitro* drug sensitivity. Of the samples genotyped, polymorphism at *Pfmrp1* codon I876V was the most frequent, with 59.3% (163/275) mutants, followed by F1390I, 7.2% (20/278), H191Y, 4.0% (6/151), and S437A, 3.3% (9/274). A significant decrease in median 50% inhibition concentrations ($IC_{50}$s) and interquartile range (IQR) was noted; AQ from 2.996 ng/ml [IQR = 2.604–4.747, n = 51] in 2008 to 1.495 ng/ml [IQR = 0.7134–3.318, n = 40] (*P*<0.001) in 2019, QN from 59.64 ng/ml [IQR = 29.88–80.89, n = 51] in 2008 to 18.10 ng/ml [IQR = 11.81–26.92, n = 42] (*P*<0.001) in 2019, CQ from 35.19 ng/ml [IQR = 16.99–71.20, n = 30] in 2008 to 6.699 ng/ml [IQR = 4.976–9.875, n = 37] (*P*<0.001)

**Data Availability Statement:** All relevant data are within the paper and its Supporting Information files.

**Funding:** Funding for this study was provided by the Armed Forces Health Surveillance Branch (AFHSB) and its Global Emerging Infections Surveillance (GEIS) Section, Grant P0209_15_KY. Additional funding provided by Schmidt Science Fellows in partnership with the Rhodes Trust. The study sponsors had no role in study design; in the collection, analysis, and interpretation of data; in the writing of the report; and in the decision to submit the paper for publication. The corresponding author should confirm that he or she had full access to all the data in the study and had final responsibility for the decision to submit for publication.

**Competing interests:** The authors have declared that no competing interests exist.

in 2019, and ART from 2.680 ng/ml [IQR = 1.608–4.857, n = 57] in 2008 to 2.105 ng/ml [IQR = 1.266–3.267, n = 47] ($P$ = 0.0012) in 2019, implying increasing parasite sensitivity to the drugs over time. However, no significant variations were observed in LU ($P$ = 0.2692) and MQ ($P$ = 0.0939) respectively, suggesting stable parasite responses over time. There was no statistical significance between the mutation at 876 and parasite sensitivity to selected antimalarials tested, suggesting stable sensitivity for the parasites with 876V mutations. These findings show that Kenyan parasite strains are still sensitive to AQ, QN, CQ, ART, LU, and MQ. Despite the presence of *Pfmrp1* mutations in parasites among the population.

## Introduction

Prior to the Corona Virus Disease 2019 (COVID-19) pandemic, substantial gains were made towards malaria eradication as disease burden had significantly reduced in the past decade. Unfortunately, the COVID-19 pandemic neutralized these gains as transmission intensity between 2020 and 2023 exceeded that of the previous years' [1, 2]. Furthermore, anticipations that malaria control and elimination were achievable were based on these positive progresses before the COVID-19 pandemic. From a baseline of 2016–2030, the Global Technical Strategy (GTS) aims to achieve a reduction of 90% of malaria morbidity incidence and mortality rate by 2030. Additionally, the strategy aims to eliminate malaria in at least 35 countries and pre-vent reintroduction in all countries that have achieved elimination [3]. While the gains to date are magnificent, the global malaria challenge remains substantial, and the rate of progress is slowing [2].

Progress towards a malaria free world is likely hindered by several present and emerging challenges. The introduction of artemisinin-based combination therapies (ACTs) has been central to achieving GTS, but changes in parasite sensitivity to these drugs in Southeast Asia (SEA) [4] and Africa [5, 6] are one of the major challenges that must be urgently and swiftly addressed. Chemotherapy has tremendously been affected worldwide by the emergence of drug resistance in *Plasmodium falciparum (P. falciparum)*, while the intense distribution of parasite strains that are resistant to chloroquine in most of the endemic areas [7] has added more complications in the treatment of malaria. Recently described genotypic and phenotypic markers are evidenced to be valid tools in monitoring artemisinin (ART) and piperaquine (PQ) resistance in SEA [8, 9]. However, these genetic tools that are relevant in monitoring altered sensitivities to ACTs in SEA might not be applicable in sub-Saharan Africa (sSA) [4, 10].

Although ACTs remain highly efficacious [4, 11], reduced parasite sensitivities to these drugs in sSA [5, 6] would be devastating, granted that this is where the brunt of the disease is felt. The most commonly used ACT in Africa, artemether-lumefantrine (AL), in recent years, studies have shown [12] selects for Single Nucleotide Polymorphisms (SNPs) in *Plasmodium falciparum chloroquine resistance transporter* (*Pfcrt*) gene, *Plasmodium falciparum multidrug resistance gene 1* (*Pfmdr1*) in sSA parasites [13–17] and *Plasmodium falciparum multidrug resistance protein 1* (*Pfmrp1*) gene [18]. Some of these genes change in mutation frequencies longitudinally can be used as surrogate indicators of the selective pressure that ACTs might be exerting on the parasite population. *P. falciparum* parasites remain under ongoing selective pressure from several anti-malarial drugs and also between *P. falciparum* genes associated with anti-malarial resistance So far, most studies have majorly concentrated on SNPs present in only five genes: *Pfcrt*, *Pfmdr1*, *Plasmodium falciparum dihydrofolate reductase* gene (*Pfdhfr*),

*Plasmodium falciparum dihydropteroate synthase* (*Pfdhps*), and *Plasmodium falciparum Kelch 13* (*Pfk13)* [19]. These are the most studied molecular markers of resistance in sSA; except the *Pfmrp1* gene [20], despite evidence of its association with susceptibility to antimalarial drugs.

*Pfmrp1* gene encodes an 1822-amino acid protein situated in the plasma membrane of the parasite, along with a member of the ATP-binding cassette (ABC) transporter superfamily [18]. It is known to influence parasite sensitivity through the efflux of glutathione, chloroquine (CQ), and quinine (QN) [21]. Its association with susceptibility to antimalarial drugs and positive parasite selection makes it a responsive target for tracking resistance [22]. Previous studies in various regions have shown the potential significance of *Pfmrp1* gene in anti-malarial drug resistance [18, 21, 22]. For instance, in Iran, four years after the introduction of ACTs, the *Pfmrp1* gene polymorphisms associated with artemisinin resistance, namely; 191Y (76.5%), 437A (69.5%), 876V (64.5%), and 1390I (17%) were found in their populations [23]. Additionally, a study conducted on the China-Myanmar border associated *Pfmrp1* SNPs at codons; N325S, H785N, T1007M, F1390I, I876V as well as H191Y, and S437A with reduced *in vitro* susceptibilities to CQ, LU, dihydroartemisinin (DHA), and piperaquine (PQ) [24]. A greater than 60%, 50%, and 7% frequency were evidenced in (H191Y and S437A), I876V and F1390I polymorphisms, respectively, in the population [24]. Furthermore, in the Thai-Myanmar border *Pfmrp1* F1390I SNPs were considerably linked to ART, MQ, and LU in an *in vitro* reduced sensitivity test in the parasite isolates [25]. Also, in Thailand and Angolan isolates, a prevalence of 95.3% *Pfmrp1* alleles 191Y and 100% 437A was shown to be associated with reduced *in vitro* drug responses to MQ, therefore confirming overexpression of the gene in other regions as well [26]. Another *in vivo* study conducted in returning travellers to Sweden from East African countries (Malawi, Uganda, Kenya, and Zanzibar) found the most prevalent non-synonymous SNPs in Africa were I876V, and K1466R. These SNPs are largely spread in Asian, African and Oceanian parasite populations. I876V was evidenced to be under selection after treatment with LU [18].

*Pfmrp1* gene mutations are known to cause reduced parasite sensitivity *in vitro* to antimalarial drugs, including ART [23], a partner drug in the combination for treatment of uncomplicated malaria in sSA [12]. Elsewhere, ACTs have been implicated in selecting for *Pfmrp1* gene polymorphisms, which impair parasite responses to antimalarial drugs [23]. Since the rollout of ACTs in Kenya, there has been limited information on the impact of this drug on the *Pfmrp1* gene. To address this gap, our study assessed temporal trends of *Pfmrp1* mutations and antimalarial sensitivity patterns of *P. falciparum* field isolates collected from four of the five malaria ecological zones in Kenya between 2008 and 2019 using genomic and *in vitro* malaria SYBR Green I assay. The outcome of this study highlights the temporal trends of *Pfmrp1* mutations and parasite sensitivity patterns to ART, LU, CQ, AQ, QN, and MQ. Further, our findings underscore the effect of *Pfmrp1* mutations on selected antimalarials sensitivity to the parasites during the widespread use of AL. This will strengthen antimalarial surveillance tools and inform malaria control policies in the region as we move into the malaria elimination phase.

## Materials and methods

### Study sites and sample collection

This study retrospectively analyzed samples collected between 2008 and 2019 under the protocol entitled, Epidemiology of Malaria Drug Sensitivity Patterns in Kenya, which was approved by the Ethical Review Committee of the Kenya Medical Research Institute (KEMRI, Protocols #1330 and #3628), Nairobi, Kenya, and the Walter Reed Army Institute of Research (WRAIR, Protocols #1384 and #2454) Institutional Review Board, Silver Spring, MD.

The samples were obtained from individuals, averagely aged six months and older, residing near participating field sites in county or sub-county hospitals, namely, Kisumu County referral hospital (KDH), Kericho County referral hospital (KCH), Kisii County referral hospital (KSI), Kombewa Sub-county hospital (KOM), Malindi Sub-county hospital (MDH), and Marigat Sub-county hospital (MGT). These sites were selected to represent four of the five malaria ecological zones in Kenya. Eligible patients presenting at outpatient departments with symptoms of malaria and/or testing positive for uncomplicated malaria by rapid diagnostic test (mRDT; Parascreen® (Pan/Pf), Zephyr Biomedicals, Verna Goa, India) were recruited into the study after providing written informed consent or assent [27, 28]. 2–3 millilitres (mls) of whole blood samples were collected for *in vitro* antimalarial drug susceptibility testing, mRDT testing, smear preparation, DNA extraction, and nucleic acid analysis, as earlier described [27]. About 100 microliters (µls) of each sample was spotted on FTA filter paper (Whatman Inc., Bound Brook, New Jersey, USA) as backup samples. The final diagnosis results were based on clinical evaluation confirmed by mRDT and/or microscopy. All *Plasmodium* positive cases were treated with AL in accordance with the Kenya Ministry of Health's recommended case management guidelines for uncomplicated malaria [29, 30]. Immediately after the blood draw, the attending clinician administered and directly observed taking of the first dose of AL based on the patient's weight. Each patient was given the remainder of the full dose of AL and advised to take the next dose after eight hours, then follow up with the remaining doses at 12-hourly intervals till completion of the dose. Further, they were encouraged to return to the hospital should symptoms persist [28]. Data on this study was accessed between 19-June-2019 and 25-February-2023.

## Genotypic analysis of *Pfmrp1* gene

Parasite DNA was extracted from whole blood using the QIAamp Blood Mini kit (Qiagen, Valencia, CA, USA) as recommended by the manufacturer [27]. The extracted DNA was stored at -20˚ C until analyzed. Genotyping of *Pfmrp1* H191Y, A437S, I876V, and F1390H alleles was determined by a PCR-based single-base extension on the Agena MassARRAY® platform (Agena Biosciences, San Diego, CA, USA) following manufacturer recommendations [27]. Primers and the assay were designed using the Agena assay design suite (ADS) (Agena Biosciences, San Diego, CA, USA). SNPs were designated as pure (which contains only either wild-type or mutant strains) or mixed (which contained both wild type and mutant alleles) based on the presence of two major peaks on the Matrix Assisted Laser Desorption Ionization-Time of Flight Mass Spectrometry (MALDI-TOF MS) or spectra as described [31]. Analysis determined the frequency of mutant alleles, where all the mixed genotypes were considered mutants.

There are several steps in performing quality assurance on the MALDI-TOF MS. Firstly, the iPLEX assay design ensures that no two alleles are within 15 Daltons of each other, thus greatly reducing any allele bias. Secondly, the MALDI-TOF platform uses a 3-point calibrant (three unique oligonucleotides of known mass) in each run that establishes an equation for the best-fit curve [32]. Lastly, quality control reactions are included in each run. This includes DNA that is known to perform well for other iPLEX reactions; a well subjected to PCR and iPLEX reaction but lacks any DNA; and a well with Taq polymerase but not subjected to the PCR stage. All these are run in duplicate and assist in evaluating background noise.

## Drugs preparation

Drug concentrations ranging from highest to lowest, for example, 3,876 to 1.89 nanomolar (nM), were made on 96-well culture plates (catalog no. 167008; Nunc, Inc., Roskilde,

Denmark) by loading the working concentration of 3,876 nM onto the first column of wells (1A to 1H), followed by 2-fold serial dilutions across 12 wells using the Biomek FXP automated laboratory workstation (Beckman Coulter, Inc., Fullerton, CA, USA) as described by Desjardins and co-workers [33]. For both culture-adapted and immediate *ex-vivo* assays, 3.12 µl of drug aliquots were made from 96-well microculture plates and transferred to 384-well plates (catalog no. 142761; Nnc, Inc.) [34–36].

## Drug susceptibility testing by the malaria SYBR Green 1 assay

The susceptibility of all field isolates alongside the control strains to six selected antimalarials, namely; AQ, QN, CQ, ART, LU, and MQ was established using the *in vitro* and *ex vivo* malaria SYBR Green 1 method [36]. The initial sample collection was conducted before the participants were treated. There were no detectable levels of AL in the serum when the sample was collected, as guided by the study protocol, since AL treatment was done after sample collection. Since no drug was present in the sample, immediate ex vivo assays were done within six hours of sample collection, while cultured samples were tested after parasite culture adaptation in the laboratory. Archived and *P. falciparum* control strains mefloquine-resistant, and chloroquine-sensitive "Indochina (3D7)", mefloquine-sensitive, and chloroquine-resistant "Indochina (DD2)", chloroquine-resistant, and mefloquine-sensitive, artemisinin "sensitive Indochina (W2)" were maintained in continuous culture as previously described [27] to attain replication robustness, rising in counts to greater than 3 to 8% infected red blood cells / total red blood cells count prior to initiation in vitro drug susceptibility tests. Further, freshly collected and culture adapted parasites were subjected to immediate *ex vivo* and *in vitro* assays as described [34, 36]. Parasite replication inhibition was quantified by measuring the per-well relative fluorescence units (RFU) of SYBR Green I dye using the Tecan Genios Plus® (Tecan US. Inc., Durham, NC), with excitation and emission wavelengths of 485 nanometres (nM) and 535 nM. The $IC_{50}$s values for the drugs were calculated using the Graph-Pad prism® software for windows® (Graph pad software, San Diego, CA, USA) as previously described [34, 36].

## Statistical analyses

SNP data was expressed as proportions, which were analyzed using the Chi-square or Cochran Armitage test for trends to establish the trends and frequencies of polymorphisms at individual codons. Drug susceptibility data was expressed as median 50% inhibition concentrations ($IC_{50}$s). Our study assessed the temporal trends of *Pfmrp1* gene mutations and the sensitivity of associated antimalarials between 2008 and 2019. The study period was categorized into three time points, namely: 2008–2009, 2013–2014, and 2018–2019. The 2008–2009 time point was categorized as the initiation/roll out phase since it took some time for effective implementation of ACTs in Kenya following a treatment policy change in 2006 [37]. From 2013 to 2014 was categorized as the transition phase while 2018 to 2019 was implementation phase and stabilization phase. Variations in median $IC_{50}$s between different study periods were examined using the non-parametric Kruskal-Wallis H test. Post-hoc analyses were done using Dunn's Multiple Comparisons test. Association between *Pfmrp1* genotype and $IC_{50}$s data was computed using the Mann-Whitney U test. All the analyses were performed using Graph pad prism version 5.0 Software® (GraphPad Software, San Diego, CA, USA). All statistical analyses were performed at the 5% significance level and the corresponding 95% confidence interval (CI). The critical significance level was set at $P \leq 0.05$.

**Table 1. Frequency of SNPs in *Pfmrp1* gene in field isolates during the study period.**

| Codons | Genotypes | Study period | | | P-value | Overall |
|---|---|---|---|---|---|---|
| | | 2008–2009 | 2013–2014 | 2018–2019 | 2008–2019 | 2008–2019 |
| | | n (%) | n (%) | n (%) | | n (%) |
| H191Y | Wild-type | 28 (100) | 71 (94.7) | 46 (95.8) | | 145 (96.0) |
| | Mutant | 0 (0) | 4 (5.3) | 2 (4.2) | 0.4661 | 6 (4.0) |
| S437A | Wild-type | 57 (100) | 100 (94.3) | 108 (97.3) | | 265 (96.7) |
| | Mutant | 0 (0) | 6 (5.7) | 3 (2.7) | 0.1396 | 9 (3.3) |
| I876V | Wild-type | 24 (41.4) | 43 (39.4) | 45 (43.2) | | 112 (40.7) |
| | Mutant | 34 (58.6) | 66 (60.6) | 63 (56.8) | 0.9402 | 163 (59.3) |
| F1390I | Wild-type | 55 (94.8) | 104 (95.4) | 99 (89.2) | | 258 (92.8) |
| | Mutant | 3 (5.2) | 5 (4.6) | 12 (10.8) | 0.1621 | 20 (7.2) |

Data is presented as frequencies [n, (%)] of Wild-type or Mutant SNPs in *P. falciparum* field isolates collected between the study periods 2008 and 2019. (n) represents the number of isolates successfully genotyped at each codon. SNPs-Single Nucleotide Polymorphisms. $P > 0.05$ shows not statistically significant proportions of mutations between study periods determined by Chi-square test.

## Results

### Prevalence of SNPs in *Pfmrp1* gene

A total of 300 archival field isolates collected from KDH, KOM, KSI, KCH, MDH, and MGT between 2008 and 2019 were genotyped for mutations in the four amino acid positions on the *Pfmrp1* gene. Genotyping findings revealed the highest prevalence of *Pfmrp1* mutations at codon 876 at 59.3% (163/275) in the field parasites during the study period. The occurrence of mutations at the remaining codons was as follows: F1390I 7.2% (20/278), H191Y 4.0% (6/151), and S437A 3.3% (9/274). Particularly, the prevalence of *Pfmrp1* mutation at codon 437 was the least reported among our parasites (Table 1). Temporal trends of mutations at the four *Pfmrp1* codons were not statistically significant between the three study periods: H191Y ($P = 0.4661$), S437A ($P = 0.1396$), I876V ($P = 0.9402$), and F1390I ($P = 0.1621$). The occurrence of mutants in different study periods is shown (Table 1).

### Drug sensitivity parasites response in the field isolates

A subset of the field isolates (n = 182) was successfully analyzed for *in vitro* drug susceptibility to AQ, QN, CQ, ART, LU, and MQ. Isolates showed diverse median $IC_{50S}$ (Table 2). The

**Table 2. Performance of selected drugs against field isolates during the study period.**

| Drugs | Median $IC_{50}s$ (ng/ml) |
|---|---|
| AQ | 2.659 [IQR = 1.383–4.161, n = 152] |
| QN | 36.13 [IQR = 15.35–66.18, n = 128] |
| CQ | 10.76 [IQR = 6.094–37.80, n = 90] |
| ART | 2.704 [IQR = 1.579–5.544, n = 152] |
| LU | 14.61 [IQR = 5.376–26.43, n = 116] |
| MQ | 5.967 [IQR = 3.166–10.13, n = 155] |

Data presented as the median 50% inhibition concentrations ($IC_{50}s$) (ng/ml), (% lower and upper interquartile ranges) in field isolates between 2008 and 2019. (n) represents isolates successfully tested. Abbreviations: CQ-chloroquine, MQ-mefloquine, QN-quinine, ART-artemisinin, AQ-amodiaquine, LU-lumefantrine.

**Table 3. *In vitro* susceptibility responses of field isolates to antimalarials during the study period.**

| Study period | 2008–2009 | | 2013–2014 | | | 2018–2019 | | |
|---|---|---|---|---|---|---|---|---|
| Drugs | Median IC50s (IQR) | n | Median IC50s (IQR) | | n | Median IC50s (IQR) | n | P-value |
| AQ | 2.996 (2.604–4.747) | 51 | 2.037 (1.387–3.936) | | 61 | 1.495 (0.7134–3.318) | 40 | <0.001* |
| QN | 59.64 (29.88–80.89) | 51 | 50.23 (32.68–78.66) | | 35 | 18.10 (11.81–26.92) | 42 | <0.001* |
| CQ | 35.19 (16.99–71.20) | 30 | 10.58 (6.105–32.63) | | 23 | 6.699 (4.976–9.875) | 37 | <0.001* |
| ART | 2.680 (1.608–4.857) | 57 | 5.114 (1.777–8.953) | | 48 | 2.105 (1.266–3.267) | 47 | 0.0012* |
| LU | 13.66 (8.278–18.74) | 27 | 11.17 (2.720–26.22) | | 53 | 19.38 (3.270–33.68) | 36 | 0.2692 |
| MQ | 7.308 (3.737–12.13) | 54 | 6.201 (3.250–10.87) | | 60 | 4.828 (3.091–7.293) | 41 | 0.0939 |

*In vitro* median 50% inhibition concentrations (IC50s) for selected antimalarials: AQ-Amodiaquine, QN-Quinine, CQ-Chloroquine, ART-Artemisinin, LU-Lumefantrine, MQ-Mefloquine in Kenyan field isolates between 2008 and 2019 study periods (2008–2009, 2013–2014, 2018–2019). (n) represents isolates successfully tested. Lower and upper interquartile ranges (IQR). *- show a statistically significant *P*-value < 0.05, determined by the Kruskal-Wallis H test; comparisons done using Dunn's Multiple Comparisons test.

reference strains response to the selected antimalarials under study is as shown in supplementary (S5 Table).

## Drug sensitivity patterns of parasites during the study period

A significant decrease in the median IC$_{50S}$ of AQ, QN, CQ, and ART was observed, from 2.996 ng/ml [IQR = 2.604–4.747, n = 51] in 2008 to 1.495 ng/ml [IQR = 0.7134–3.318, n = 40] (*P*<0.001) in 2019, 59.64 ng/ml [IQR = 29.88–80.89, n = 51] in 2008 to 18.10 ng/ml [IQR = 11.81–26.92, n = 42] (*P*<0.001) in 2019, 35.19 ng/ml [IQR = 16.99–71.20, n = 30] in 2008 to 6.699 ng/ml [IQR = 4.976–9.875, n = 37] (*P*<0.001) in 2019, 2.680 ng/ml [IQR = 1.608–4.857, n = 57] in 2008 to 2.105 ng/ml [IQR = 1.266–3.267, n = 47] (*P* = 0.0012) in 2019, respectively, implying increasing parasite sensitivity to the drugs over time (Tables 3, 4 and Fig 1A–1D). LU median IC$_{50S}$ increased by almost two folds from 11.17 (8.273–18.740) ng/ml in 2013–2014 time point to 19.38 (3.270–33.680) ng/ml in 2018–2019 time point,

**Table 4. *In vitro* 95% CI lower and upper mean susceptibility responses of field isolates to antimalarials.**

| | 95% CI of mean | 2008–2009 | 2013–2014 | 2018–2019 |
|---|---|---|---|---|
| AQ | Lower | 3.272 | 2.339 | 1.741 |
| | Upper | 5.113 | 5.124 | 4.573 |
| QN | Lower | 50.36 | 43.17 | 15.69 |
| | Upper | 90.33 | 102.8 | 32.34 |
| CQ | Lower | 32.81 | 10.09 | 6.87 |
| | Upper | 89.48 | 25.06 | 24.93 |
| ART | Lower | 3.092 | 5.215 | 2.018 |
| | Upper | 5.474 | 10.79 | 2.826 |
| LU | Lower | 11.62 | 11.52 | 14.94 |
| | Upper | 17.63 | 19.09 | 25.63 |
| MQ | Lower | 7.861 | 6.424 | 4.607 |
| | Upper | 19.03 | 13.19 | 7.95 |

In vitro 95% CI (Percent Confidence Interval) lower and upper mean susceptibility. In vitro median 50% inhibition concentrations (IC50s) for selected antimalarials: AQ-Amodiaquine, QN-Quinine, CQ-Chloroquine, ART-Artemisinin, LU-Lumefantrine, MQ-Mefloquine in Kenyan field isolates between 2008 and 2019 study periods (2008–2009, 2013–2014, 2018–2019).

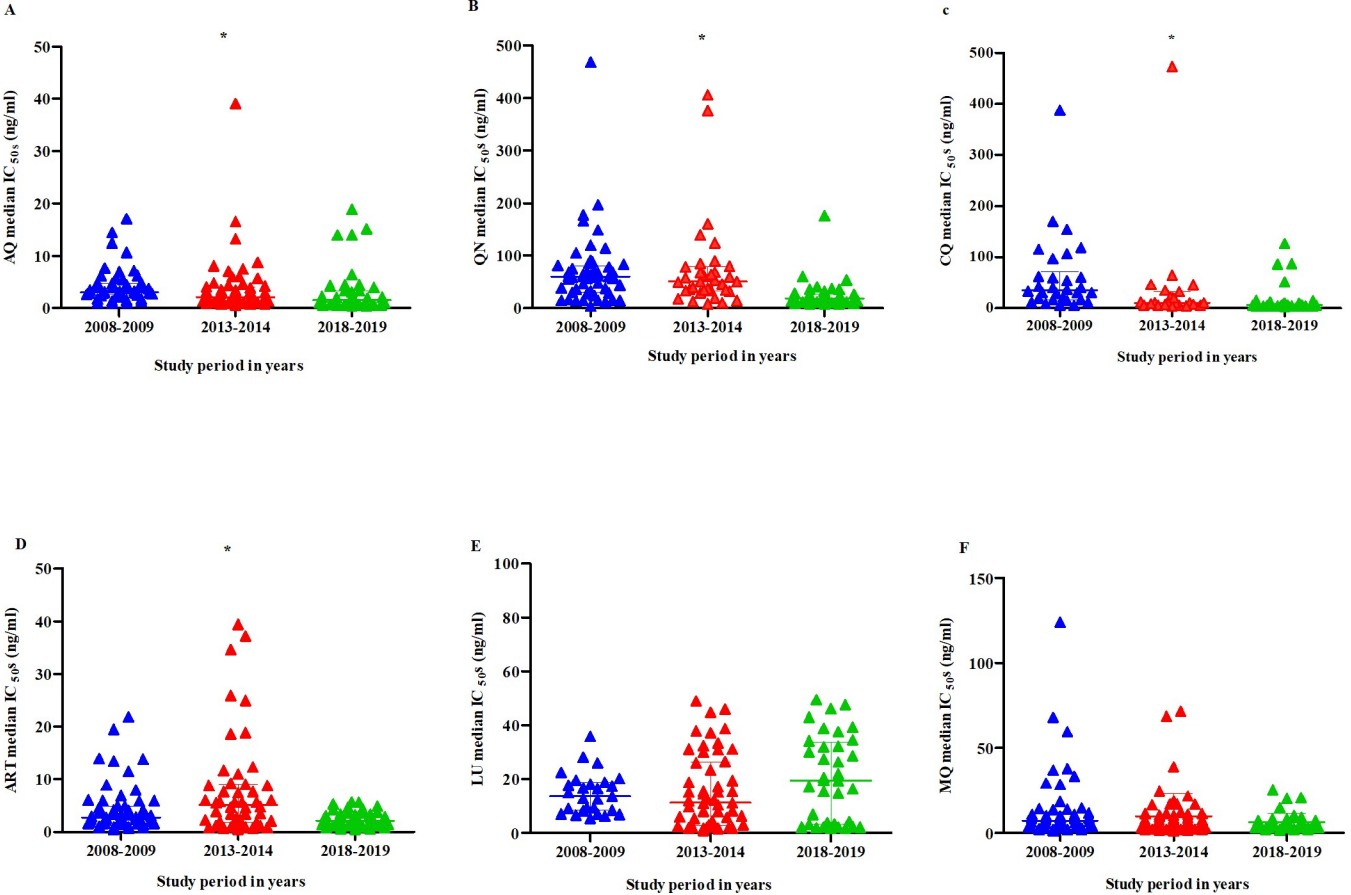

**Fig 1. Performance of selected antimalarials against field isolates during the study period.** A-F Scatter plots showing temporal variation in selected antimalarial drugs performance against the parasites between 2008–2009, 2013–2014, and 2018–2019 study periods. Horizontal bars and whiskers represent median 50% inhibition concentrations ($IC_{50}$s) and interquartile range [IQR] (ng/ml). *- show statistically significant $P$-value $< 0.05$ determined by the Kruskal-Wallis H test; comparisons done using Dunn's Multiple comparisons test.

although this was not statistically significant ($P = 0.2692$). On the contrary, no significant variations were observed in LU ($P = 0.2692$) and MQ ($P = 0.0939$) susceptibility, respectively, suggesting stable parasite responses over time (Table 3 and Fig 1E and 1F).

## Association between *Pfmrp1* polymorphisms and *in vitro* antimalarial drug susceptibility at codon 876

Analysis of the association between *Pfmrp1* polymorphisms and *in vitro* antimalarial drug sensitivity as determined by the Mann-Whitney U test revealed no significant relationships between the *Pfmrp1* I876V and the selected antimalarials tested (Fig 2A and 2B). The mutant genotype (876V) at this codon had no effect on the $IC_{50}$s of the drugs under study, suggesting stable sensitivity of the parasites ($P>0.05$) (Fig 2A and 2B).

## Discussion

This study established three key findings: Firstly, genotyping findings revealed the presence of *Pfmrp1* I876V, F1390I, H191Y, and S437A mutations in field isolates at different frequencies. Secondly, a significant decrease in the median $IC_{50S}$ of AQ, QN, CQ, and ART was noted, implying increasing parasite sensitivity to the drugs over time while sensitivity to LU and MQ

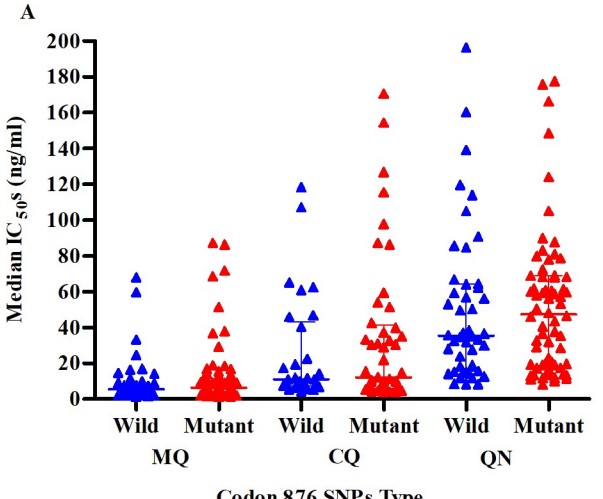

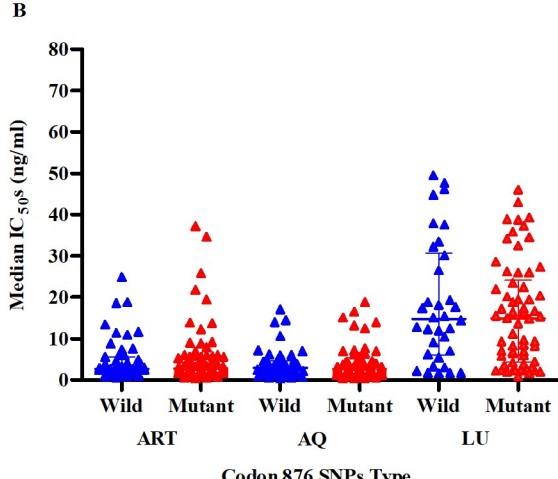

**Fig 2. Association of polymorphic IC$_{50}$s versus wild-type at codon 876 in selected drugs tested.** A-B Scatter plots showing association between drugs median 50% inhibition concentrations (IC$_{50}$s) and parasite genotypes determined by Mann-Whitney U test. *- show statistically significant *P*-value < 0.05. Horizontal bars and whiskers represent median IC$_{50}$s and interquartile range (IQR). MQ-Mefloquine, CQ-Chloroquine, QN-Quinine, ART-Artemisinin, AQ-Amodiaquine, LU-Lumefantrine. SNPs-Single nucleotide polymorphisms.

remained stable during the six-year observation period. Thirdly, association studies between *Pfmrp1* mutations and *in vitro* antimalarial drug sensitivity revealed no significant change in IC$_{50}$s in parasites carrying *Pfmrp1* 876V mutation, suggesting stable sensitivity of these parasites to the antimalarials. Monitoring the frequency of mutations associated with antimalarial resistance over a prolonged period can unveil patterns in allele selection in a population with time and can expand the remedial life of the present and upcoming cures [38]. The emergence of drug resistance can either be identified in time by *in vitro* susceptibility testing, changing temporal and spatial patterns of parasite drug sensitivity, or changes in the responses of individual drugs currently used as well [38]. It is on that account, that we tracked resistance by estimating the frequency of resistance-linked mutations in the *Pfmrp1* gene and *in vitro* susceptibility testing in Kenyan isolates for better surveillance of ACTs resistance in the population over time.

Important tools for conserving ACT efficacy are to keep track of selection and respond to emerging indicators of drug resistance [39]. The *Pfmrp1* SNPs are already established and well documented in the Greater Mekong Sub-region (GMS) and are associated with *in vivo* and *in vitro* responses to ACTs [18, 25]. Analysis of a sample from a traveller returning to Sweden from Kenya in 2009, only detected 876V mutation in the *Pfmrp1* gene [18]. Our assessment of temporal trends in the frequency of SNPs in the *Pfmrp1* gene between 2008 and 2019 in Kenya during the ACTs rollout phase, showed the highest frequency of *Pfmrp1* gene mutations at codon I876V, followed by F1390I, H191Y, and S437A (Table 1). However, 191Y, 437A, 876V, and 1390I mutations occurred in Kenyan isolates at lower frequencies, which were not significantly varying between study periods as compared to what was earlier reported in Iran, Thai-Myanmar border, and Myanmar townships and areas [23, 25, 39, 40], revealing the diversity in parasite evolution across regions. Based on earlier findings, AL has been shown to select for CQ-sensitive parasites [13]. High frequency of 876V mutations noted correspondingly in Iran, Thai-Myanmar border, Myanmar populations [17, 22, 41] might be due to the use of the current treatment regimen AL against CQ-sensitive parasites. African parasites have harboured low frequencies of *Pfmrp1* SNPs previously in the four loci compared to Asia and Oceania

parasites [18, 23]. This could be attributed to the high transmission intensities in Africa compared to Oceania and Asia [18], since low transmission intensities favour the spread of drug resistance and high levels of clinical immunity create a natural ecological refuge for drug-sensitive parasites [42].

Drug susceptibility assays during the same study period showed a significant decrease in the median $IC_{50}$s of CQ, AQ, QN, and ART over the 2008 and 2019 study periods. Concurrently, our reports on increasing CQ sensitivity in the Kenyan parasite strains were in line with what was observed in our recent and earlier studies [27, 34, 43] and in the French Guyana study between 2008 and 2012 [44]. This observation in our parasite population is due to the withdrawal of CQ drug pressure from the parasites at the onset of sulfadoxine-pyrimethamine (SP) use in Kenya [45] and later (in 2008), the introduction of AL as the first-line treatment for malaria [5, 45, 46]. Increasing QN sensitivity corroborates findings from GMS [46]. Results in both populations on QN may be attributed to 1390I polymorphisms in *Pfmrp1* gene, which are associated with QN improved sensitivity [22]. Furthermore, the observation of a corresponding increase in CQ and QN sensitivity in this study is contrary to the findings reported earlier in the Northeast Myanmar and China-Myanmar border region where declining cases of *in vitro* CQ sensitivity was marked by increasing QN sensitivity [22, 47]. The reason for the persistent evolution of CQ resistance in GMS could be as a result of continued selection pressure due to its use for treating sympatric *Plasmodium vivax* malaria [48]. AQ has been used both as monotherapy and combined therapy with SP for the treatment of uncomplicated malaria in Africa [49, 50]. Our *in vitro* analysis of Kenyan parasite strains response to AQ is suggestive of increasing sensitivity to this drug. This concurs with a study in Senegal between 2013 and 2015, where increasing activity of CQ and monodesethylamodiaquine, the active metabolite of AQ, was noted [51]. Conversely, a significant increase in CQ and AQ was reported in Kenyan isolates between 2005 and 2013 [43, 52]. This is suggestive of a reciprocate reduction in the use of AQ and withdrawal of CQ for malaria treatment during the study period and reduced drug pressure that selects for *Pfcrt* gene, the notable marker for AQ and CQ drug resistance [43].

Our ART findings contradicted those of an earlier study conducted in the same population [52]. In addition to previous studies in SEA and East Africa (Rwanda and Uganda) that reported declining sensitivity of ART after adoption of ACTs [4, 6, 40, 53–56]. Moreover, a four-year assessment of the sensitivity of Nigerian isolates to ART ten years following the deployment of ACTs depicted decreasing susceptibility to ART [57]. ART declining sensitivity in SEA and the mentioned East African regions is linked to *Kelch13* polymorphisms in the parasites causing partial artemisinin resistance in those populations [4, 6, 40, 53–56]. Despite our contrary findings of increased ART sensitivity, continued surveillance for early detection of resistance in Kenya, where it forms part of the current treatment regimen, is still encouraged [52].

In our study, LU and MQ displayed a stable sensitivity against the parasites between 2008 and 2019 study period.

LU is a partner drug to artemether, which forms the first-line regimen in most countries in sSA [43], although previous studies in Kenya and Africa show declining LU sensitivity as a result of continued use of AL [27, 58, 59]. Our study findings show a stable response of the Kenyan parasite strains to LU despite a declining parasite sensitivity to this drug being depicted between 2013–2014 and 2018–2019 study periods. Our previous study by Wakoli and colleagues assessing the impact of parasite genomic dynamics on the sensitivity *of P. falciparum* to PQ and other antimalarial drugs between 2008 and 2021 showed declining sensitivity to LU [27]. Our study outcome is different from the earlier finding because this study analyzed samples from a short study period (2008–2019), hence the small sample size; thus, further investigation using a large sample size should be conducted to elucidate this. MQ was

previously used as a monotherapy for the treatment of uncomplicated malaria in Thailand, but after four years of use, resistant parasites emerged, rendering it ineffective [41]. However, cases of resistance against MQ have continued to be reported on the Thai-Myanmar border [41]. Our observation of sustained MQ sensitivity against Kenyan parasite strains during the study period is because it has been infrequently used for the treatment of uncomplicated malaria due to a perceived poor tolerance, and this shows its potential to be used as a frontline antimalarial in the country [60].

Studies in Africa and Thai-Myanmar border areas have shown that SNPs in *Pfmrp1* gene at codons 876 and 1390 are strongly associated with reduced MQ, ART, QN, and LU susceptibility [18, 22, 25]. Even though the other two mutations at codons 191 and 437 have been associated with reduced CQ and QN sensitivity [18, 22]. In our association study between these mutations and the *in vitro* activity of the drugs under investigation, there was no significant association between the median $IC_{50}$s of the studied antimalarials and the 876V mutation. This contradicts earlier findings of significant associated reductions in sensitivity to the antimalarials studied [18, 22, 25]. However, this association was contrary to other studies conducted on parasite isolates from the same region [41]. Additionally, studies in the North-East and China-Myanmar border associated 876V mutations with reduced *in vitro* susceptibility to CQ and AL in African isolates [22, 24, 55]. Hence, further investigations are warranted with larger sample sizes for proper conclusions in Kenya, where this mutation is also prevalent. Uniquely, when three samples per drug with the least and highest $IC_{50}$s were sampled, 876V mutations were exhibited in nearly all the samples with both high and low sensitivities. This may be attributed to selection pressure spreading these mutations in the parasites on the population, and it thus needs to be further investigated if it is a negative or positive selection associated with the hitchhiking effect.

This study had some limitations: Firstly, the sample size from some study sites was small, and three study sites (KOM, MGT, and MDH) were not opened during the 2008 and 2009 base years. Secondly, we were unable to culture-adapt samples, especially from some field sites, mostly due to logical challenges, and therefore only a subset of 182 out of 300 samples were successfully tested and analyzed. Lastly, apart from codon 876, we did not gather enough data from the other three codons (191, 437, and 1390) to generate discernible relationships with $IC_{50}$s, probably due to low frequencies.

## Conclusion

Our study findings denote that Kenyan *P. falciparum* parasite strains are sensitive to AQ, QN, CQ, ART, LU, and MQ despite the detection of *Pfmrp1* mutations in the parasite population.

## Supporting information

**S1 Table. Primers used for analysis of four SNPs in one gene.** The primers in (S1 Table) were generated by the design software, guided by Agena MassARRAY® system. 4 SNPs were designed into 3 pools. The primary polymerase chain reaction (PCR) that is locus-specific PCR was amplified with pools of 1st PCRP and 2nd PCRP for specific SNP loci. This secondary PCR uses mass-modified dideoxy nucleotide terminator of an oligonucleotide primer (UEP_-SEQ). Primer anneals immediately upstream of the polymorphic site of interest. The SNPs were added in a multiplexed single base pair extension (SBE) with dideoxy nucleotides that are mass modified depending on the allele and design of the assay. The extended primers were then detected by MALDI-TOF MS in the analyzer.
(XLSX)

**S2 Table. Number of samples per site during the study period.** Summary of the number of samples analyzed from each study site per study period.
(XLSX)

**S3 Table. Drugs $IC_{50}$s data during the study period.** Outliers in blue font excluded during Kruskal-Wallis analysis. Scatter plots were graphed and outliers excluded from the ones outside the correct ranges. Correct ranges are values in between the first third wells and the last third wells after serial dilutions as per drug starting concentration. Starting concentrations in (ng/mls); ART-200, QN-4000, CQ-2000, LU-200, MQ-500, AQ-200.
(XLSX)

**S4 Table. SNPs $IC_{50}$s data at codon 876 during the study period.** Outliers in blue font excluded during Mann-Whitney analysis. Scatter plots were graphed and outliers excluded from the ones outside the correct ranges. Correct ranges are values in between the First third wells and the last third wells after serial dilutions as per drug starting concentration. Starting concentrations in (ng/mls); ART-200, QN-4000, CQ-2000, LU-200, MQ-500, AQ-200.
(XLSX)

**S5 Table. $IC_{50}$s data of antimalarials against controls.** The sensitivity data of control strains to the selected antimalarial drugs under study are shown (XLSX). Susceptibility of W2, DD2 and 3D7 control strains to AQ, ART, QN, LU, MQ and CQ.
(XLSX)

**S6 Table. *Pfmrp1* gene SNPs data during the study period.** Wild-type genotype versus mutant mentioned per codon. Mixed genotypes were categorized as mutants as well in frequencies.
(XLSX)

## Acknowledgments

We would like to thank the patients who participated in this study, clinical team, and other support staff at the study sites. We are grateful to our colleagues in the Malaria Drug Resistance laboratory for their technical and moral support. We would also like to thank the Director KEMRI for permission to publish this work. The opinions and assertions contained herein are the private opinions of the authors and are not to be construed as official or as reflecting the views of the U.S. Department of Defense or the Walter Reed Army Institute of Research.

## Author Contributions

**Conceptualization:** Winnie Okore, Collins Ouma, Luicer O. Ingasia, Duncan M. Wakoli, Joseph G. Amwoma, Ben Andagalu, Hoseah M. Akala.

**Data curation:** Raphael O. Okoth, Redemptah Yeda, Luicer O. Ingasia, Edwin W. Mwakio, Douglas O. Ochora, Duncan M. Wakoli, Joseph G. Amwoma, Gladys C. Chemwor, Jackline A. Juma, Charles O. Okudo, Agnes C. Cheruiyot, Benjamin H. Opot, Dennis Juma, Ben Andagalu, Amanda Roth, Edwin Kamau.

**Formal analysis:** Winnie Okore, Collins Ouma, Raphael O. Okoth, Redemptah Yeda, Luicer O. Ingasia, Edwin W. Mwakio, Douglas O. Ochora, Duncan M. Wakoli, Joseph G. Amwoma, Jackline A. Juma, Agnes C. Cheruiyot, Benjamin H. Opot, Dennis Juma, Timothy E. Egbo, Ben Andagalu, Edwin Kamau, Hoseah M. Akala.

**Funding acquisition:** Luicer O. Ingasia, Ben Andagalu, Amanda Roth, Hoseah M. Akala.

**Investigation:** Winnie Okore, Collins Ouma, Joseph G. Amwoma, Dennis Juma, Edwin Kamau, Hoseah M. Akala.

**Methodology:** Winnie Okore, Raphael O. Okoth, Redemptah Yeda, Edwin W. Mwakio, Douglas O. Ochora, Duncan M. Wakoli, Joseph G. Amwoma, Gladys C. Chemwor, Jackline A. Juma, Charles O. Okudo, Agnes C. Cheruiyot, Dennis Juma, Hoseah M. Akala.

**Project administration:** Collins Ouma, Dennis Juma, Hoseah M. Akala.

**Resources:** Luicer O. Ingasia, Ben Andagalu, Amanda Roth, Edwin Kamau, Hoseah M. Akala.

**Software:** Winnie Okore, Raphael O. Okoth, Redemptah Yeda, Luicer O. Ingasia, Edwin W. Mwakio, Joseph G. Amwoma, Benjamin H. Opot.

**Supervision:** Collins Ouma, Benjamin H. Opot, Dennis Juma, Ben Andagalu, Edwin Kamau, Hoseah M. Akala.

**Validation:** Winnie Okore, Collins Ouma, Raphael O. Okoth, Douglas O. Ochora, Duncan M. Wakoli, Gladys C. Chemwor, Dennis Juma, Timothy E. Egbo, Hoseah M. Akala.

**Visualization:** Winnie Okore.

**Writing – original draft:** Winnie Okore, Luicer O. Ingasia, Edwin W. Mwakio, Duncan M. Wakoli, Joseph G. Amwoma.

**Writing – review & editing:** Collins Ouma, Raphael O. Okoth, Redemptah Yeda, Edwin W. Mwakio, Douglas O. Ochora, Gladys C. Chemwor, Agnes C. Cheruiyot, Benjamin H. Opot, Dennis Juma, Timothy E. Egbo, Ben Andagalu, Amanda Roth, Edwin Kamau, Hoseah M. Akala.

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
