## [Decision Letter · Decision Letter 0]

26 Sep 2023

PONE-D-23-23753Temporal trends of Plasmodium falciparum multi-drug resistance protein 1 gene mutations during implementation of artemisinin combination therapies in Kenya between 2008 and 2019PLOS ONE

Dear Dr. Akala, 

Thank you for submitting your manuscript to PLOS ONE. After careful consideration, we feel that it has merit but does not fully meet PLOS ONE’s publication criteria as it currently stands. Therefore, we invite you to submit a revised version of the manuscript that addresses the points raised during the review process.

We look forward to receiving your revised manuscript.

Kind regards,

Moses Ocan, PhD

Academic Editor

PLOS ONE

 [Funding for this study was provided by the Armed Forces Health Surveillance Branch (AFHSB) and its Global Emerging Infections Surveillance (GEIS) Section, Grant P0209_15_KY. Additional funding provided by Schmidt Science Fellows in partnership with the Rhodes Trust. The study sponsors had no role in study design; in the collection, analysis, and interpretation of data; in the writing of the report; and in the decision to submit the paper for publication. The corresponding author should confirm that he or she had full access to all the data in the study and had final responsibility for the decision to submit for publication.]

Additional Editor Comments:

The authors should additionally, address the following questions.

1) Artemether and Lumefantrine, are they active antimalarial compounds or their metabolites? please justify their use in In vitro assays in this study

2) Why is CQ IC50 lower than that of QN? provide justification and how quality control was ensured during the assays

3) Provide more information on the implications of the findings of this study on use of Artemisinin based therapies (ACTs) and CQ for malaria treatment.

Reviewers' comments:

Reviewer's Responses to Questions

**Comments to the Author**

1. Is the manuscript technically sound, and do the data support the conclusions?

Reviewer #1: Yes

Reviewer #2: Partly

2. Has the statistical analysis been performed appropriately and rigorously? 

Reviewer #1: Yes

Reviewer #2: Yes

3. Have the authors made all data underlying the findings in their manuscript fully available?

Reviewer #1: Yes

Reviewer #2: No

4. Is the manuscript presented in an intelligible fashion and written in standard English?

Reviewer #1: Yes

Reviewer #2: No

5. Review Comments to the Author

Reviewer #1: The manuscript tackles a good topic in Antimalarial drug resistance and its relevant because it looks at a gene that is rarely explored. The authors did the standard experiments for checking for in vitro and ex vivo sensitivities. I had a few concerns when it came to these assays

1. In vitro and ex vivo assays were not differentiated in the manuscript. As we know, during culture adaptation, some clones (in my experience most Lumefantrine tolerant strains), tend to be lost during culture, so these samples should be separated in the analysis.

2. In the abstract, they give changes in the IC50s of the drugs, but they do not attach the changes to a specific drug, so its hard for a reader to know which drugs had the changes.

3. In the statistical analysis, they mention, they used chi-square test to test for the trends, I think the authors used the chi-square test for trends also known as the Cochran-Armitage trend test. This should be very clear in the methods, because for most analyses the typical chi-square test could not be used in most of those analyses.

4. I think the authors ignored a very interesting finding. The lumefantrine IC50s increased to 19.38ng/ml in 2018-2019 from 11.17 in 2013-2014 and 2008-2009. Even though this trend p-value was not significant, there was almost a 2 fold increase in the IC50s. In my opinion, this should be discussed as something of concern, but stating that this was not statically significant, given the fact that most upcoming literature is showing the same trend especially in areas where AL is being used.

5. In the discussion, the authors describe why they chose the 3 time points, in my opinion this should be in the methods giving a rationale for picking these 3 time points, other than going to the discussion section

Reviewer #2: Summary: Winnie Okore et al intended to identify if there are mutation (SNPs) in Plasmodium falciparum and if these mutations confer resistance to common antimalaria in use among Kenya populations. They also set out to determine if malaria parasites are still sensitive to these anti-malarial agents. They used samples from a primary longitudinal study with objective of monitoring drug resistance and malaria epidemiology among Kenyan populations.

Comments

Title does not depict the key message to the public health perspective. Something like “Increased sensitivity of malaria parasite strains to common antimalarial drugs in use by Kenyan populations” would be more captivating and appealing to policy makers, public health gurus, pharmaceuticals, and other non-biomedical scientists.

Abstract

Grammatical and composition issues

…have previously been reported to confer resistance to Artemisinin and its partner drugs in Southeast Asia…what do you mean by “its partner drug”…In my opinion, I would put it like…”have previously been reported to confer resistance to Artemisinin based Combination Therapies (ACTs). I am not comfortable with the statement..”assayed for SNPs”….assayed should be used when your test methods give quantitative details, but in your case, you simply detected presence or absence of the SNPs…would propose something like…tested for the presence of SNPs.

The sentence starting with.. “Nonetheless” can be replaced by “However”…There is mismatch in statement “mutation at 876 were not significantly associated with” .. I suggest either “mutation at 876 did not significantly affect the performance of selected antimalarial at IC50S” or “there was no statistical significance between mutation at 876 and parasite sensitivity to selected antimalarial. The statement “Kenyan parasites are sensitive” is ambiguous … The parasites are not made in Kenya or restricted to Kenyan population. A statement like “malaria parasites among Kenyan populations are still sensitive to…. Despite the presence of Pfmrp1 in parasites among the population. ” would be more befitting.

Technical issues: before you conclude that there is increased or even just sustained sensitivity, kindly clarify the following issues.

1)is the composition of the active substance in those selected drugs the same for the entire study period?; If yes, how did you prove/measure it?

2)Were the drugs procured from the same Manufacturer or vendor for the entire study period?

3)There are so many confounding that can lead to that outcome, ranging from patient’s factors, parasites factors, environmental factors, therapeutics, technical competency factors among others. How did you control for these variables/factors in your study?. Without rigorously controlling for these factors, you cannot confidently conclude that there was increased sensitivity.

Introduction

First paragraph is mixed up and colloquial….It should read something like; prior to COVID19 pandemic, substantial gains were made towards malaria eradication as disease burden had significantly reduced in the past decade. Unfortunately, COVID19 pandemic neutralized these gains as transmission intensity between 2020 to 2023 exceeded that of the previous years….So many sentences are hanging, words have been misplaced or misused among others. I suggest the entire introduction should be re-written in standard grammar and English composition.

Materials and methods

Study sites & sample collections:

The paragraph starting with..”This retrospective study…” I guess should be like “This study retrospectively analyzed samples collected between 2008 and 2019…the statement “under the protocol”, Is this the name of the primary study or you actually meant the primary study which your current study was nested in?. On Ethical approval, you indicated that this study was approved by KEMRI REC and Walter Reed. Was this ethical approval for your study or for the primary study?, If it was for your study, why was Walter Reed involved in the approval when the entire study was executed in Kenya?

Use more suitable joining words, adjectives and adverbs in the paragraph which start with “the samples were obtained” up to “marigat sub-county hospital” . The word “capture” may be replaced by “represent” .

From your submission, inclusion criteria were through a positive RDT and presentation with malaria related symptoms. If yes, who screened for these symptoms and what were those symptoms, what happened to those with severe malaria?... I guess these are details of the primary study which I think has been published. To save you from all these questions, cite the study and refer the readers there for details. You used the abbreviation “AL”, Is it a standard abbreviation in use and what does it mean since you did not define it in the abstract?. That dosing of AL (I guess stands for Artemisinin/Lumefantrine) of 8 hours and followed by 12 hours, is that the standard in the Kenyan Clinical guideline for treatment of uncomplicated malaria, kindly be sure?

Genotypic analysis of Pfmrp1

What do you mean by “genomic DNA”?, is there another type of DNA for Plasmodium falciparum?..just putting parasite DNA would be sufficient. Put a proper negative sign where you write -20. Also degree centigrade should be oc, not oC as you put it. Were the primers designed in house or commercially procured?. How did you perform quality assurance of matrix assisted laser desorption ionization-Time of flight mass spectrometry?, Kindly include a section on its Quality assurance. Are there instances on the MALTI-TOF MS platform where you can obtain pure SNPs which are all mutants as you have put in your characterization ?

Drug preparation:

The first paragraph is confusing. I guess you meant.. Drug/disc concentration ranging from …..to….. was serially diluted in a 96 well culture plate. What was the dilution factor and why the factor?

Susceptibility testing

I find so many missing links in this section which I think is the core of the lab assay. You indicated earlier that all these patients were treated with Artemisinin/lumefantrine, but then you are testing the sensitivity of the samples from these patients against six other anti-malarial?, isn’t there any antagonistic, synergistic, or additive effects between AL and these drugs in the experiment?, What is the T1/2 of AL once in the serum?. After administration of A/L, how long did you wait before setting up the experiment for the other drugs?. Did you culture only control strains or even parasites from the field?. How did you ensure quality of results from SYBR Green dye and Tecan Genios?. Whilst the authors referred us to already published protocols or studies, I feel there is need to add more information to this section.

Statistical analysis:

What basis did you use to categorize your isolates into those years? Does Mann-Whitney U test measure association or correlation?. Why did you choose this test over Wilcoxon signed rank test?

Results:

Tables should be labelled below not above. Could there be any reasons why the number of mutants in 1876v are higher than that of wild types for all the study points?

On drug sensitivity patterns, what was the reference standard for sensitivity and who set it?. For table 3, add a column of confidence intervals. The result has not been appropriately communicated despite the enormous experiments.

Discussion:

What do you mean by “post ACT error”?... Are ACTs being phased out in Kenya?. What do you mean by.. “can expand the remedial life of the current and upcoming cures?”. Statement “returning from Kenya to Sweden” sounds inaccurate; how about we put it like.. “Returning to Sweden from Kenya”.. the destination has to come before the origin. When you say “samples from a traveler”; was this one person bled multiple times or it was just a grammatical error?. What does SEA mean as you used in the discussion? As an immunologist, I have taken it for Schizont Egress Antigen. Why would selection pressure affect mutation at 876V only and not other alleles or loci as you seemed to be concluding?. The stated limitations are so strong that it can weaken the statistical analysis and conclusions from this study.

6. PLOS authors have the option to publish the peer review history of their article (what does this mean?). If published, this will include your full peer review and any attached files.

Reviewer #1: No

Reviewer #2: No

---

## [Author Response · Author response to Decision Letter 0]

27 Nov 2023

Temporal trends of Plasmodium falciparum multi-drug resistance protein 1 gene mutations during implementation of artemisinin combination therapies in Kenya between 2008 and 2019

Please submit your revised manuscript by Nov 10 2023 11:59PM. 

 [Funding for this study was provided by the Armed Forces Health Surveillance Branch (AFHSB) and its Global Emerging Infections Surveillance (GEIS) Section, Grant P0117_22_KY. The study sponsors had no role in study design; in the collection, analysis, and interpretation of data; in the writing of the report; and in the decision to submit the paper for publication. The corresponding author should confirm that he or she had full access to all the data in the study and had final responsibility for the decision to submit for publication.] 

Please include this amended Role of Funder statement in your cover letter; we will change the online submission form on your behalf

Response: Added in the cover letter 

Response: Thank you for the observation. The minimal data set underlying the results described in our manuscript can be found as Supporting Information files (S3-6).

Response: well noted. No ethical no legal restrictions 

Response: Thank you for the comment. This observation is well noted. Our ethics statement is included in the manuscript methods section under the first paragraph of study sites and sample collection section page 6 line 214-217 and has also been deleted after the acknowledgement section.

Response: Reference list checked one by one to ensure it was complete and correct.

List also updated from 27 to 59. References 28, 34 and 53 added.

Additional Editor Comments:

The authors should additionally, address the following questions.

7. Artemether and Lumefantrine, are they active antimalarial compounds or their metabolites? 

Thank you for the comment. Both artemether and lumefantrine are not the active metabolites but were tested as such. The active metabolite of Artemether is dihydroartemisinin while that of lumefantrine is desbutyl-lumefantrine. A study by Salman and coworkers show that a combined desbutyl-lumefantrine and lumefantrine AUC0–∞ weighted on in vitro antimalarial activity was inversely associated with recurrent parasitemia, suggesting that both the parent drug and the metabolite contribute to the treatment outcome of artemether-lumefantrine. (Salman et al 2011). 

ii) please justify their use in in vitro assays in this study

Thank you for the comment. Resistance to antimalarial drugs including lumefantrine in Southeast Asia (SEA), Oceania and Africa have been associated with polymorphisms in Pfmrp1 codons; 191, 437, 876, 1390. The mutations at the 4 codons are already established and validated non-synonymous SNPs frequent in the populations and are associated with resistance to the chosen drugs. Evidence of positive selection on the Pfmrp1 gene and its association with reduced sensitivities of parasites to the selected antimalarials was based on the following studies (Dahlström et al., 2009; Gupta et al., 2014b; Pirahmadi, Zakeri, Afsharpad, & Djadid, 2013; Veiga et al., 2011; Zhao et al., 2019). However, in Kenya this genotypic and the impact on parasite susceptibility to selected antimalarials is scanty. Therefore, this study was conducted to fill this gap to inform policies on malaria prevention interventions. These molecules were assayed for in vitro because other studies have previously reported their in vitro readouts in different transmission from those for this study. Similarly, this study intended to measure the response of isolated from natural infections occurring in the study area so as to provide parallel results.

8. Why is CQ IC50 lower than that of QN? provide justification and how quality control was ensured during the assays

Response: Thank you for the comment. This observation in our parasite population is due to the withdrawal of CQ drug pressure from the parasites at the onset of sulfadoxine-pyrimethamine (SP) use in Kenya (Balikagala et al., 2020) and increased use of AL (Mwai et al., 2009; Balikagala et al., 2020) which has further selected for parasites that are susceptible to chloroquine. However, QN is still used to date in the treatment of severe malaria. The quality control IC50s threshold of QN is also a bit higher in QN as compared to CQ. Reference standard for sensitivity was carried out by testing of reference strains with well characterized profiles for example Dd2 (Mefloquine sensitive), D6, 3D7 clone is sensitive to chloroquine drug with standard cut off of <10ng/ml and < 45ng/ml in field isolates while W2 is resistant to chloroquine with cut off value of >45ng/ml. Mefloquine cut off in field isolates < 10ng/ml and in W2 clone. Quinine cut off is <275ng/ml. Also the starting concentrations in the drug testing for quinine is high, the concentration ranges (ng/mL), CQ (1,000 to 1.953), and QN (2,000 to 3.906), because chloroquine is more potent than quinine. This finding are in line with what has been reported previously (Zhao et al., 2022). 

9. Provide more information on the implications of the findings of this study on use of Artemisinin based therapies (ACTs) and CQ for malaria treatment. 

a. Response: Thank you for the comment. Our study findings show that malaria parasites among Kenyan populations are still sensitive to AQ, QN, CQ, ART, LU and MQ. Despite the presence of Pfmrp1 mutations in parasites among the population. Based on the selected antimalarials comprising ACTs, they have proven to be still effective for malaria treatment in Kenya. CQ can also be possibly reused in future since its sensitivity has improved over time following the release of drug pressure on the parasites.

Review Comments to the Author

Reviewer #1: The manuscript tackles a good topic in Antimalarial drug resistance and its relevant because it looks at a gene that is rarely explored. The authors did the standard experiments for checking for in vitro and ex vivo sensitivities. I had a few concerns when it came to these assays

1. In vitro and ex vivo assays were not differentiated in the manuscript. As we know, during culture adaptation, some clones (in my experience most Lumefantrine tolerant strains), tend to be lost during culture, so these samples should be separated in the analysis. 

Thank you for the comment. Our analysis did not separate ex vivo and in vitro samples because in previous studies isolates subjected to ex vivo and in vitro assays have shown marginal difference IC50 (Akala et al., 2011), meaning there is no difference in terms of parasite response to antimalarials in both assays. Based on these findings, all our subsequent studies on drug susceptibility assays have analyzed these samples using the approach adopted in this study (Akala et al., 2011, Eyase et al., 2013, Wakoli et al., 2022). 

2. In the abstract, they give changes in the IC50s of the drugs, but they do not attach the changes to a specific drug, so its hard for a reader to know which drugs had the changes.

Response: Thank you for the comment. This section has been revised as follows for easy readability. The section reads, “A significant decrease in median 50% inhibition concentrations (IC50s) and interquartile range (IQR) was noted; AQ from 2.996 ng/ml [IQR=2.604-4.747, n=51] in 2008 to 1.495 ng/ml [IQR=0.7134-3.318, n=40] (P<0.001) in 2019, QN from 59.64 ng/ml [IQR=29.88-80.89, n=51] in 2008 to 18.10 ng/ml [IQR=11.81-26.92, n=42] (P<0.001) in 2019, CQ from 35.19 ng/ml [IQR=16.99-71.20, n=30] in 2008 to 6.699 ng/ml [IQR=4.976-9.875, n=37] (P<0.001) in 2019 and ART from 2.680 ng/ml [IQR=1.608-4.857, n=57] in 2008 to 2.105 ng/ml [IQR=1.266-3.267, n=47] (P=0.0012) in 2019, implying increasing parasite sensitivity to the drugs over time. Abstract section, Page 1, line 58-63.”

3. In the statistical analysis, they mention, they used chi-square test to test for the trends, I think the authors used the chi-square test for trends also known as the Cochran-Armitage trend test. This should be very clear in the methods, because for most analyses the typical chi-square test could not be used in most of those analyses.

Response: Thank you for your observation. The chi-square test for trends is also called the Cochran –Armitage test for trends. Chi-square/ Cochran Armitage test for trends was used to establish the frequencies and trends of polymorphisms at individual codons in the methods under statistical analyses section, Page 10, Line 302.

4. I think the authors ignored a very interesting finding. The lumefantrine IC50s increased to 19.38ng/ml in 2018-2019 from 11.17 in 2013-2014 and 2008-2009. Even though this trend p-value was not significant, there was almost a 2 fold increase in the IC50s. In my opinion, this should be discussed as something of concern, but stating that this was not statically significant, given the fact that most upcoming literature is showing the same trend especially in areas where AL is being used. 

Response. Concur. Thank you for the comment.This has been included un the manuscriptin line 436 to 440. The section reads, “). LU median IC50S increased by almost two folds from 11.17 (8.273-18.740) ng/ml in 2013-2014 time point to 19.38 (3.270-33.680) ng/ml in 2018-2019 time point, although this was not statistically significant (P=0.2692).” 

5. In the discussion, the authors describe why they chose the 3 time points, in my opinion this should be in the methods giving a rationale for picking these 3 time points, other than going to the discussion section

Response: Thank you for your observation. This is a great suggestion. The description has been moved to the methods section under statistical analyses Page 10-11, line 304-316 and removed from the discussion section. 

Reviewer #2: Summary: Winnie Okore et al intended to identify if there are mutation (SNPs) in Plasmodium falciparum and if these mutations confer resistance to common antimalaria in use among Kenya populations. They also set out to determine if malaria parasites are still sensitive to these anti-malarial agents. They used samples from a primary longitudinal study with objective of monitoring drug resistance and malaria epidemiology among Kenyan populations.

Comments

1. Title does not depict the key message to the public health perspective. Something like “Increased sensitivity of malaria parasite strains to common antimalarial drugs in use by Kenyan populations” would be more captivating and appealing to policy makers, public health gurus, pharmaceuticals, and other non-biomedical scientists. Dr. Akala/LTC Kamau

Response: concur that the current title is silent on the implication. The reason is that clinical results from tiral need to accompany the report for us to go for the proposed title. If we go for it now, reviewers will return to that it is an overtone, on the account of the threshold of non in vivo work 

2. Abstract

Grammatical and composition issues

…have previously been reported to confer resistance to Artemisinin and its partner drugs in Southeast Asia…what do you mean by “its partner drug”…In my opinion, I would put it like…”have previously been reported to confer resistance to Artemisinin based Combination Therapies (ACTs).

Response: Thank you for the comment. This is a great suggestion. The statement by its partner drug has been replaced with “Artemisinin based Combination Therapies (ACTs). Abstract section, page 1, line 52.

a. I am not comfortable with the statement..”assayed for SNPs”….assayed should be used when your test methods give quantitative details, but in your case, you simply detected presence or absence of the SNPs…would propose something like…tested for the presence of SNPs. 

Response: Thank you for the comment. This is a great suggestion. The statement “a

---

## [Decision Letter · Decision Letter 1]

29 Jan 2024

Temporal trends of Plasmodium falciparum multi-drug resistance protein 1 gene mutations during implementation of artemisinin combination therapies in Kenya between 2008 and 2019

PONE-D-23-23753R1

Dear Dr. Akala,

We’re pleased to inform you that your manuscript has been judged scientifically suitable for publication and will be formally accepted for publication once it meets all outstanding technical requirements.

Kind regards,

Jesse N. Gitaka, M.D.

Academic Editor

PLOS ONE

Additional Editor Comments (optional):

Reviewers' comments:

Reviewer's Responses to Questions

**Comments to the Author**

1. If the authors have adequately addressed your comments raised in a previous round of review and you feel that this manuscript is now acceptable for publication, you may indicate that here to bypass the “Comments to the Author” section, enter your conflict of interest statement in the “Confidential to Editor” section, and submit your "Accept" recommendation.

Reviewer #2: (No Response)

2. Is the manuscript technically sound, and do the data support the conclusions?

Reviewer #2: Yes

3. Has the statistical analysis been performed appropriately and rigorously? 

Reviewer #2: Yes

4. Have the authors made all data underlying the findings in their manuscript fully available?

Reviewer #2: Yes

5. Is the manuscript presented in an intelligible fashion and written in standard English?

Reviewer #2: No

6. Review Comments to the Author

Reviewer #2: I note that the quality of the manuscript has greatly improved. The quality of the English composition and grammar has significantly improved. There is now a logical flow in the manuscript and it seems there is now a technical input in every section. Results section has been expounded to reflect the enormous work done in the laboratory

I still have a few concerns though

Under authorship? Who is the corresponding author for this paper? Is it Winnie or Hosea?.. because both have put in their email addresses and Hosea has done most of the responses, but from the authorship list, Winnie seems to be the corresponding author.

Methods

1) statement " two folds serial dilutions" can be interpreted differently. How about you mention the exact dilution factor...say 1 in 80?

2) descriptions under drug preparations are not adding up and sentences seems to be hanging, especially sentence from "both culture adapted"

3) statement " There were no traces" sounds analytically inaccurate... It is analytically difficult to prove absolute absence of a drug or any biomolecule in blood samples of such a huge non controlled population. Non committal statements like.."There were no detectable levels of" fits better.

Results

The 95% CI for CQ and QN seems very wide compared to the rest of the drugs. Could there be any plausible explanations/justification?

Discussion

1) I still have a problem with the words used in the categorization of the study period. I am uncomfortable with using the word "era". There are better synonyms like " study phase" or just " study period". I can tolerate the word "era" but not "post ACT era". You clearly responded that ACTs are still actively in use in Kenya and so "post ACT era" does not apply here". If I were you, I would just categorize the study periods into "Initiation/roll out phase, transition/implementation phase and stabilization phase".

2) I see you have maintained the statement " Kenyan parasite" through out your write up. May be use "Kenyan parasite strain". At least a strain can be localized within a geographical region since your study had the geo-spatial component.

3) Is it the drug which is sensitive to the parasite or it is the parasite which is sensitive to the drug?.. I believe it is the later (parasite sensitive to the drug). Kindly read through the entire manuscript and ensure every relevant statement states so.

4) Is the Country "French Guiana" or French Guyana? (Line 397)

5)Was the use of the word "sympatric" intentional or you meant " symptomatic".

Conclusion

1) I still find a conclusion that a two fold increase being statistically insignificant to be very difficult. Perhaps it is because of the wide confidence interval as earlier stated.

General comment: I know there are English literature experts in Kenya (At least we used to enjoy the KBC English news bulletin from the neighboring country during childhood). Let one of them go through the revised manuscript just to align the grammar, English composition, punctuation, etc. Once these few minor comments are addressed, I am approving the manuscript to proceed to the next stage.

7. PLOS authors have the option to publish the peer review history of their article (what does this mean?). If published, this will include your full peer review and any attached files.

Reviewer #2: No

---

## [Editor Report · Acceptance letter]

9 Jun 2024

PONE-D-23-23753R1 

PLOS ONE

Dear Dr. Akala, 

I'm pleased to inform you that your manuscript has been deemed suitable for publication in PLOS ONE. Congratulations! Your manuscript is now being handed over to our production team.

Kind regards, 

on behalf of

Dr. Jesse Gitaka 

Academic Editor

PLOS ONE